# Changes in Disposition of Ezetimibe and Its Active Metabolites Induced by Impaired Hepatic Function: The Influence of Enzyme and Transporter Activities

**DOI:** 10.3390/pharmaceutics14122743

**Published:** 2022-12-08

**Authors:** Ningjie Xie, Hong Wang, Hua Qin, Zitao Guo, Hao Xue, Jiafeng Hu, Xiaoyan Chen

**Affiliations:** 1University of Chinese Academy of Sciences, No. 19A Yuquan Road, Beijing 100049, China; 2Shanghai Institute of Materia Medica, Chinese Academy of Sciences, 501 Haike Road, Shanghai 201203, China; 3Beijing Tianheng Pharmaceutical Research Institute Co., Ltd., No. 93, Zone 6, Doudian Village, Beijing 102433, China

**Keywords:** hepatic impairment, ezetimibe, active metabolites, phase II enzymes, transporters, first-pass metabolism, inhibitory bile acids

## Abstract

Ezetimibe (EZE) is a selective cholesterol absorption inhibitor. Hepatic impairment significantly increases the systemic exposure of EZE and its main active phenolic glucuronide, EZE-Ph. Although changes in efflux transporter activity partly explain the changes in EZE-Ph pharmacokinetics, the causes of the changes to EZE and the effects of the administration route on EZE-Ph remain unclear. A carbon tetrachloride (CCl_4_)-induced hepatic failure rat model was combined with in vitro experiments to explore altered EZE and EZE-Ph disposition caused by hepatic impairment. The plasma exposure of EZE and EZE-Ph increased by 11.1- and 4.4-fold in CCl_4_-induced rats following an oral administration of 10 mg/kg EZE, and by 2.1- and 16.4-fold after an intravenous injection. The conversion of EZE to EZE-Ph decreased concentration-dependently in CCl_4_-induced rat liver S9 fractions, but no change was observed in the intestinal metabolism. EZE-Ph was a substrate for multiple efflux and uptake transporters, unlike EZE. In contrast to efflux transporters, no difference was seen in the hepatic uptake of EZE-Ph between control and CCl_4_-induced rats. However, bile acids that accumulated due to liver injury inhibited the uptake of EZE-Ph by organic anion transporting polypeptides (OATPs) (glycochenodeoxycholic acid and taurochenodeoxycholic acid had IC_50_ values of 15.1 and 7.94 μM in OATP1B3-overexpressed cells). In conclusion, the increased plasma exposure of the parent drug EZE during hepatic dysfunction was attributed to decreased hepatic glucuronide conjugation, whereas the increased exposure of the metabolite EZE-Ph was mainly related to transporter activity, particularly the inhibitory effects of bile acids on OATPs after oral administration.

## 1. Introduction

Chronic liver disease (CLD) is a complex, progressive disease that causes over one million deaths annually and is the tenth most common cause of death worldwide [1]. The hepatitis B virus, the hepatitis C virus, nonalcoholic fatty liver disease (NAFLD), and alcohol-associated liver disease are common causes of CLD [2,3]. Regardless of the cause, repeated bouts can lead to an inflammatory response, matrix deposition, parenchymal cell death, and angiogenesis, which can develop into progressive fibrosis [4] and eventually result in liver cirrhosis, hepatocellular carcinoma, and death [5].

The liver is the major site of drug metabolism and it determines key pharmacokinetic characteristics, including the rates of liver extraction, liver clearance, and plasma protein binding. It is well documented that liver injury alters drug absorption and disposition, thereby affecting drug efficacy and safety. Naloxone, an antagonist to opioids, is used for the long-term treatment of opioid addiction. However, its clearance decreases by over 50% and the plasma exposure increases by three-fold in patients with moderate CLD, while the plasma concentration increases by over ten-fold in patients with severe CLD [6]. Therefore, this drug is contraindicated for individuals with severe CLD and a lower usage is recommended for patients with moderate CLD. Some clinical drugs, such as clopidogrel and cyclophosphamide, possess no intrinsic pharmacological activity, but generate active metabolites under the action of liver enzymes. However, liver disease can reduce the activity of some enzymes and hinder the formation of active metabolites. Therefore, caution is advised in the use of these drugs by CLD patients [7]. Consequently, there is a clinical need to investigate the effects and mechanisms of hepatic impairment on the disposition of drugs in order to optimize medications for individual patients.

Several studies have reported that the occurrence of CLD in individuals can alter the expression and function of drug-metabolizing enzymes and transporters. However, the results reported in the literature are slightly different. For example, CYP2E1 activity was unchanged during the progression of NAFLD [8], whereas its activity significantly decreased in patients with moderate–severe liver disease [9]. In addition, the protein abundance of the bile salt export pump (BSEP) remained unchanged in CLD patients (Child–Pugh class C), whereas the BSEP expression decreased in NAFLD patients [10].

Liver fibrosis is an important pathological characteristic of CLD [11]. Carbon tetrachloride (CCl_4_) has been widely used as a chemical inducer of experimental liver fibrosis [12,13,14]. Similar to CLD patients, the activity of hepatic CYP1A2, CYP2E1, and CYP3A decreased in CCl_4_-induced rats [15]. In addition, in vivo studies revealed that the transport ability of multidrug resistance-associated protein (MRP) 2 decreased, whereas BSEP was unchanged [16]. Therefore, CCl_4_-induced rats are often used as an animal model to investigate the significance of hepatic dysfunction on the pharmacokinetics of CLD patients [12,17,18,19,20].

Liver disease disrupts bile acid homeostasis, and bile acids have been used as biomarkers of liver injury for many years. Compared with healthy subjects, the concentration of the serum total bile acids (TBA) in CLD patients increased by 4.24- to 50.3-fold, while the conjugated bile acid concentrations changed more than those of the unconjugated bile acids. Luo et al. reported that cholic acid (CA) was 3.1-fold higher in CLD patients, whereas its taurine conjugated form, taurocholic acid (TCA), was elevated by 97.2-fold [21]. Bile acids are substrates of UDP-glucuronosyltransferases (UGTs) and transporters, such as UGT1A3, UGT2B4, UGT2B7, sodium (Na^+^)-taurocholate cotransporting polypeptide (NTCP), organic anion transporting polypeptides (OATPs), and MRPs [22,23,24,25]. Although not yet reported, it is theorized that the pharmacokinetics of some drugs and metabolites subject to these enzymes and/or transporters may be affected by bile acids accumulating due to liver injury.

Ezetimibe (EZE), 1-(4-fluorophenyl)-3(R)-[3-(4-fluorophenyl)-3(S)-hydroxypropyl]-4(S)-(4-hydroxyphenyl)-2-azetidinone, is a selective cholesterol absorption inhibitor that acts primarily by inhibiting the sterol uptake transporter Niemann–Pick C1-like 1 protein at the villus tip of the enterocytes of the small intestine [26]. Following oral administration, EZE undergoes an intense intestinal first-pass metabolism to predominantly form a phenolic glucuronide (ezetimibe phenoxy β-D-glucuronide, EZE-Ph) and, to a lesser extent, a benzylic glucuronide (ezetimibe hydroxy β-D-glucuronide, EZE-Hy) (Figure 1) [27]. EZE-Ph is more potent than EZE for inhibiting cholesterol absorption [28]. The parent and conjugated metabolites (EZE-Ph and EZE-Hy) account for approximately 10–20% and 80–90% of the EZE-related material in human plasma, respectively [29]. It has been reported that multiple transporters are involved in the pharmacokinetics of EZE conjugates, consisting of OATPs, multidrug resistance protein 1 (MDR1), and MRP2 [30,31]. However, the parent drug was only a weak substrate for MDR1 and MRP2 [31].

Hepatic impairment has significant effects on the pharmacokinetics of EZE and its conjugated metabolites. The plasma exposure to EZE was 1.4-, 5.8-, and 4.8-fold higher in mild, moderate, and severe CLD patients than in healthy subjects, while the plasma exposure to conjugated metabolites was 1.7-, 3.1-, and 4.0- fold higher, respectively [32]. Hardwick et al. [33] investigated the altered disposition of glucuronide metabolites in a nonalcoholic steatohepatitis (NASH) rat model, which is similar to humans. A combination of induced expression and the altered localization of efflux transporters in NASH rats resulted in elevated plasma exposure to glucuronide conjugates. However, the reason for the elevated exposure to EZE following hepatic dysfunction is unknown. Furthermore, our previous study demonstrated that the influence of liver injury on the pharmacokinetics of EZE and EZE-Ph differed between oral and intravenous administrations.

To elucidate the underlying mechanisms of the elevated plasma exposures to EZE and its glucuronide conjugates in CLD patients, this study constructed a CCl_4_-induced hepatic failure rat model, combined with in vitro experiments, to explore the role of metabolizing enzymes and transporters in the altered disposition of EZE and its metabolites. The effect of disrupted bile acid homeostasis on the enzymes and transporters involved in the disposition of EZE and its metabolites was also investigated.

## 2. Materials and Methods

### 2.1. Chemicals and Reagents

EZE (99.8% purity) was kindly provided by the Tianheng Pharmaceutical Research Institute Co., Ltd. (Beijing, China). EZE-Ph (98.0% purity), EZE-Hy (95.0% purity), and ezetimibe-d4 (EZE-d4, 98.0% purity) were purchased from Toronto Research Chemicals, Inc. (North York, ON, Canada). CA, chenodeoxycholic acid (CDCA), ursodeoxycholic acid (UDCA), glycocholic acid (GCA), glycochenodeoxycholic acid (GCDCA), glycoursodeoxycholic acid (GUDCA), TCA, taurochenodeoxycholic acid (TCDCA), and tauroursodeoxycholic acid (TUDCA) were obtained from Yuanye Bio-Technology Co., Ltd. (Shanghai, China). TRIzol regent, Dulbecco’s modified Eagle’s medium, Williams’ E medium, insulin–transferrin–selenium, fetal bovine serum, GlutaMAX-I, nonessential amino acids, trypsin, penicillin G, and streptomycin were purchased from Invitrogen (Carlsbad, CA, USA). Uridine 5′-diphosphoglucuronic acid (UDPGA), alamethicin, and Krebs–Henseleit buffer were purchased from Sigma-Aldrich (St. Louis, MO, USA). Tris-HCl, Hanks’ balanced salt solution (HBSS), DNase/RNase-free distilled water, and a BCA protein assay kit were purchased from Meilun Biology Technology (Dalian, China). Plates (24-well, 48-well, or 96-well) biocoated with collagen I or poly-D-lysine were provided by BD Biosciences (San Jose, CA, USA). Alanine aminotransferase (ALT), aspartate aminotransferase (AST), and alkaline phosphatase (AKP) assay kits were sourced from Nanjing Jiancheng Bioengineering Institute (Jiangsu, China). PrimeScript^TM^ RT Master Mix kit and TB Green^®^ Premix Ex Taq^TM^ II kit were provided by Takara Biomedical Technology Co., Ltd. (Osaka, Japan). Purified water was obtained via a Millipore Milli-Q gradient water purification system (Molsheim, France). All other commercially available solvents and regents were supplied by Sinopharm Chemical Reagent Co., Ltd. (Shanghai, China).

Human MDR1-, MRP2-, and MRP3-expressing membrane vesicles isolated from Sf9 insect cells were sourced from GenoMembrane Co., Ltd. (Yokohama, Japan).

### 2.2. Construction of CCl_4_-Induced Hepatic Failure Rat Model

All animal studies were carried out according to the Guide for the Care and Use of Laboratory Animals of the Shanghai Institute of Materia Medica, Chinese Academy of Sciences. Male Sprague–Dawley (SD) rats weighing 180 to 220 g were randomized into two groups. The CCl_4_-induced model group was administered an intraperitoneal injection of CCl_4_ (20% in corn oil) at a dose of 2.5 mL/kg three times a week for 8 weeks. The control group was treated intraperitoneally with the same volume of corn oil at the same time intervals. Blood samples were collected and centrifuged at 11,600 *g* for 5 min to obtain serum samples for the determination of serum ALT, AST, and AKP.

### 2.3. Biochemistry and Histopathologic Study

Serum ALT, AST, and AKP were measured using the corresponding assay kits following the manufacturer’s instructions. Tissues fixed in 4% paraformaldehyde were embedded in paraffin, cut into slices with a 3 μm thickness, and stained with hematoxylin and eosin (H&E). The H&E sections were observed with optical microscopy to study morphological changes.

### 2.4. Pharmacokinetic Experiments

Animals were fasted for 12 h with free access to water before the experiments. EZE was administered orally (prepared in 0.5% carboxymethylcellulose sodium) or intravenously (dissolved in 10% dimethyl sulfoxide, 10% tween-80, and 80% sterile H_2_O) to CCl_4_-induced rats (*n* = 5) and control rats (*n* = 5) at 10 mg/kg. Blood samples were collected from the retro-orbital venous plexus before administration (0 h), at 5, 15, and 30 min, and at 1, 2, 4, 6, 8, 12, 24, and 48 h after administration in tubes containing EDTA-2K. Plasma samples were harvested by centrifugation at 11,600 *g* for 5 min at 4 °C and then stored at −80 °C until further analysis.

To investigate changes due to the first-pass metabolism and biliary excretion during liver disease, CCl_4_-induced rats (*n* = 5) and control rats (*n* = 5) were anesthetized and the hepatic portal and femoral veins were cannulated with PE-50 polyethylene tubing. EZE was orally administered to rats at 10 mg/kg. Blood samples were collected simultaneously from the hepatic portal and femoral veins before administration (0 h), at 5, 15, and 30 min, and at 1, 2, 4, 6, 8, 12, 24, and 48 h after administration. Plasma samples were harvested as previously described.

After the pharmacokinetic studies, the rats were anesthetized for the excision of the duodenum, jejunum, ileum, colon, and liver. Except one part of the liver, which was fixed in 4% paraformaldehyde for the H&E stain, all tissues were put into liquid nitrogen immediately and then stored at −80 °C.

### 2.5. Liver S9 Fraction Isolation

One portion of the rat liver tissue was homogenized in a 5-fold volume of ice-cold phosphate buffer (including 250 mM sucrose, 10 mM Tris-HCl, and 1 mM EDTA-2Na, pH 7.4) and centrifugated at 9000 *g* for 20 min. After centrifugation, the supernatant was collected as the S9 fraction and stored at −80 °C. The protein content was measured using the BCA protein assay kit.

### 2.6. Human and Rat Liver S9 Fraction Incubation

To evaluate the effect of hepatic impairment on the hepatic metabolism of EZE, 0.3 μM, 3 μM, and 30 μM EZE were incubated with rat liver S9 fractions (2 mg of protein/mL) at 37 °C for 60 min in the Tris-HCl-buffered system (100 mM, pH 7.4), consisting of UDPGA (1 mM), MgCl_2_ (10 mM), and alamethicin (25 μg/mL). To investigate the effect of bile acids on the metabolism of EZE in the liver, 3 μM EZE was incubated with rat or human liver S9 fractions (2 mg of protein/mL) at 37 °C for 60 min in the Tris-HCl-buffered system with or without bile acids at 1, 10, and 100 μM. The total volume of the incubation system was 100 μL. The reactions were terminated by adding 100 μL of ice-cold acetonitrile. Samples were stored at −20 °C until analysis. EZE and EZE-Ph were detected by liquid chromatography with tandem mass spectrometry (LC-MS/MS).

### 2.7. Hepatocyte Isolation and Transporter-Mediated Uptake Assays

Rat primary hepatocytes were isolated from male SD rats as previously described [34]. Fresh rat hepatocytes (120,000 per well) were cultured in Williams’ E medium (containing 0.1 μM dexamethasone, 5% fetal bovine serum, 1% penicillin G/streptomycin, 1% GlutaMAX-I, and 1% insulin–transferrin–selenium) on collagen I-coated 48-well plates. Rat hepatocytes cultured for 4 h under a carbogen atmosphere were directly used for uptake assays.

The hepatocytes were rinsed three times with HBSS (37 °C) and equilibrated in HBSS for 10 min. The uptake assays were initiated by adding HBSS containing 10 μM EZE or EZE-Ph, with or without rifampicin (positive inhibitor, 200 μM) or bile acids (1, 10, and 100 μM). After 10 min of incubation, cells were quickly washed three times with ice-cold HBSS and then lysed by the addition of 300 μL of purified water. The protein content was measured using the BCA protein assay kit. EZE and EZE-Ph were analyzed by LC-MS/MS.

### 2.8. Liver Slices

Liver slices were prepared as previously described elsewhere [35]. Rats were anesthetized and sacrificed via exsanguination from the abdominal aorta, and the livers were excised. Liver cylindrical cores of an 8 mm diameter were made before slicing. Liver slices with a thickness of 300 μm were obtained using a Krumdieck tissue slicer MD6000 (TSE systems, Chesterfield, MO) with ice-cold and carbogen-saturated Krebs–Henseleit buffer.

After 5 min of preincubation at 4 and 37 °C, the liver slices were transferred to 24-well plates containing 500 μL of buffer with 10 μM EZE-Ph for 10 min. After incubation, the liver slices were washed 3 times with ice-cold HBSS and dried on filter paper. Each liver slice was homogenized with 200 μL of purified water and the protein content was determined using a BCA protein assay kit. The concentration of EZE-Ph was determined using LC-MS/MS.

### 2.9. Cell Culture and Uptake Studies Using Transporter-Expressing HEK293 Cells

Human OATP1B1-, OATP1B3-, OATP2B1-, OAT1-, OAT3-, or OCT2-transfected HEK293 cells and vector-transfected control cells (mock) were constructed at HD Biosciences Co., Ltd. (Shanghai, China). The functions of these transporters were evaluated using relevant positive substrates and inhibitors. Estradiol-17β-glucuronide (5 μM, OATP1B1 and OATP1B3), estrone 3-sulfate sodium salt (E3S, 5 μM, OATP2B1), para-aminohippuric acid (PAH, 20 μM, OAT1), furosemide (5 μM, OAT3), and 1-methyl-4-phenyl pyridinium (MPP^+^, 5 μM, OCT2) were used as substrates for the transporters. In addition, the inhibitor for each transporter was selected as follows: rifampicin (200 μM, OATP1B1, OATP1B3, and OATP2B1), probenecid (200 μM, OAT1, and OAT3), and TPA (200 μM, OCT2). The cells (50,000 per well) were cultured in Dulbecco’s modified Eagle’s medium (containing 10% fetal bovine serum, 1% GlutaMAX-I, 1% penicillin G/streptomycin, and 1% nonessential amino acids) on poly-D-lysine-coated 96-well plates at 37 °C in an atmosphere of 5% CO_2_. The uptake studies were carried out 48 h after seeding.

The cells were rinsed three times with HBSS (37 °C) and then equilibrated in HBSS for 10 min. The uptake assay was initiated by adding HBSS containing EZE-Ph, with or without rifampicin (positive inhibitor) or bile acids. After incubation for the designated time, the cells were quickly washed three times with ice-cold HBSS and then lysed by the addition of 100 μL of purified water. The protein content was measured using a BCA protein assay kit. The concentrations of EZE and EZE-Ph were determined using LC-MS/MS.

### 2.10. mRNA Analysis

RNA was extracted from intestine and liver tissues using the TRIzol reagent as previously described [36]. We synthesized cDNA from 500 ng of total RNA by using a PrimeScript™ RT Master Mix kit. The reaction was performed in a volume of 20 μL, containing 0.5 ng/μL of cDNA, 400 nM corresponding primer, and 10 μL of TB Green Premix Ex Taq II on an Applied Biosystems QuantStudio 5 Real-Time PCR system (Applied Biosystems, Foster City, CA, USA). The quantitative PCR conditions were 95 °C for 30 s, 95 °C for 3 s, and 60 °C for 30 s for 40 cycles. The sequences of primers were as follows: *Mdr1a* (forward) 5′-CAACCAGCATTCTCCATAATA-5′, (reverse) 5′-CCCAAGGATCAGGAACAATA-3′; *Mdr1b* (forward) 5′-CCTCCTTGGTCCTCTCAA-3′, (reverse) 5′-TGTTTGGGGCTAAATGTC-3′; *Mrp2* (forward) 5′-GCACATGGCTCCTGGTTTTG-3′, (reverse) 5′-ATACGCCGCATAAGACCGAG-3′; *Mrp3* (forward) 5′-TGTGGGTCTTTCCGTGTC-3′, (reverse) 5′-GCCTCAGTCTCCGTCTTAG-3′; and β-actin (forward) 5′-GCCACCAGTTCGCCAT-3′, (reverse) 5′-CATACCCACCATCACACC-3′. The PCR products were analyzed using ΔΔCt with β-actin as the internal standard.

### 2.11. Transport Studies with Human MDR1-, MRP2-, and MRP3-Expressing Membrane Vesicles

In transport studies, a modified rapid filtration technique was used according to the manufacturer’s protocol (GenoMembrane Co.). The uptake of test compounds (10 μM EZE or EZE-Ph) into membrane vesicles (100 μg protein/100 μL) was conducted with or without 4 mM ATP in the transport medium (50 μL, pH 7.4), which included 50 mM MOPS-Tris, 70 mM KCl, and 7.5 mM MgCl_2_. The vesicular transport was terminated after 10 min by adding 200 μL of ice-cold stop buffer containing 40 mM MOPS-Tris and 70 mM KCl. The stopped incubate was quickly transferred to a 96-well glass fiber filter plate (Millipore, MA, USA) and washed five times with ice-cold stop buffer. The EZE and EZE-Ph inside the vesicles were released by adding 200 μL of methanol–water (70:30, *v*/*v*) and analyzed by LC-MS/MS. The ATP-dependent transport was evaluated based on the ratio of transport with ATP to that without ATP.

### 2.12. Determination of EZE, EZE-Ph, and EZE-Hy

Chromatographic separation of the analytes from the matrix was achieved via a Shimadzu LC-30AD high-performance liquid chromatography system (Kyoto, Japan) on a BEH C18 column (50 × 2.1 mm i.d., 1.7 μm, Waters Corp., Milford, MA, USA) with temperature maintained at 40 °C. The mobile phase was a mixture of 0.2% formic acid in water (A) and acetonitrile (B) at a flow rate of 0.55 mL/min. The gradient elution was initiated at 20% B, maintained for the first 0.5 min, increased to 60% B linearly in 5.0 min, maintained for 0.5 min, and finally decreased to 20% B to equilibrate the column for 0.8 min. MS detection was conducted by an API 5500 triple-quadrupole MS (Applied Biosystems, ON, Canada) under electrospray ionization conditions in negative mode. The instrument parameters were as follows: ion spray voltage, −4500 V; source temperature, 550 °C; nebulizer gas, 60 psi; heater gas, 60 psi; and curtain gas, 40 psi. The mass transitions for multiple reaction monitoring were m/z 408.2 → 271.0 for EZE, m/z 584.4 → 271.0 for EZE-Ph and EZE-Hy, and 412.2 → 271.0 for Ezetimibe-d4 (internal standard). Data were acquired and processed with the Analyst 1.6.3 software (Applied Biosystems). The standard curve ranges were 1.00 to 1000 ng/mL for EZE, 3.00 to 3000 ng/mL for EZE-Ph, and 1.00 to 1000 ng/mL for EZE-Hy in plasma. The standard curve ranges were 1.00 to 1000 nM for EZE, and 1.00 to 1000 nM for EZE-Ph in cell lysates and liver S9 fractions. The intra- and inter-day precision and accuracy, the stability, the matrix effects, and the dilution integrity for EZE and its glucuronide metabolites from the quality control (QC) samples are summarized in the Appendix A.

A 25.0 μL aliquot of each sample and 25.0 μL of internal standard (50.0 ng/mL EZE-d4) were mixed with 100 μL of acetonitrile. After the samples were vortexed and centrifugated at 2250 g for 15 min, the supernatants were used to measure EZE and its metabolites by LC-MS/MS.

### 2.13. Data Analysis

WinNonlin version 6.1 (Pharsight Corp., Cary, NC, USA) was used to calculate the pharmacokinetic parameters in a non-compartmental model. Statistical comparisons between the two groups were evaluated using a Student’s two-tailed unpaired *t*-test or the nonparametric Mann–Whitney test in GraphPad Prism (version 8.0, GraphPad Software Inc, SanDiego, CA, USA). A value of *p* < 0.05 was considered significant. All data were presented as the mean ± S.D. (*n* ≥ 3).

The rates of substrate uptake were normalized using the protein contents of cell lysates. The net uptake via transporters was calculated by subtracting the uptake in vector-transfected HEK293 cells (or uptake into hepatocytes at 4 °C) from that in transporter-transfected HEK293 cells (or uptake into hepatocytes at 37 °C).

The kinetic parameters and the half inhibitory concentration (IC_50_) were calculated using GraphPad Prism. The kinetic data were analyzed by nonlinear regression fits according to the following equations: the Michaelis–Menten Equation (1), allosteric sigmoidal modeling (2), or substrate inhibition modeling (3):V = V_max_ × S/(K_m_ + S) (1)
V = V_max_ × S^h^/(K_m_^h^ + S) (2)
V = V_max_ × S/[K_m_ + S × (1 + S/K_i_)](3)
where V (pmol/min/mg protein) is the velocity of substrate uptake, S (µmol/L) is the substrate concentration, K_m_ (µmol/L) is the substrate concentration at the half-maximal uptake rate (V_max_), h is the Hill slope, and K_i_ is the inhibition constant. The intrinsic clearance (CL_int_) was calculated as CL_int_ = V_max_/K_m_.

The IC_50_ values were estimated through nonlinear regression analysis by plotting the log value of the inhibitor concentration against the normalized residual activity (%), following the equation Y (%) = 100/[1 + 10^(X−LogIC50)^].

## 3. Results

### 3.1. Biochemistry Parameters and Histopathologic Sections of Control and CCl_4_-Induced Rats

After 8 weeks of treatment, the liver index (ratio of liver weight to body weight) of the CCl_4_-induced rats was significantly higher than that of the control rats (Table 1). The serum biochemical parameters ALT, AST, and AKP were 8.62-, 5.76-, and 3.86-fold higher in the CCl_4_-induced rats than in the control rats, respectively. Figure 2 shows the histopathology results using hematoxylin–eosin staining. The liver lobules of the control rats exhibited a classical structure, with hepatocyte plates directed from the portal triads toward the central vein, where they freely anastomosed. Irregularly dilated liver sinusoids and the space of Disse were also observed. By contrast, rats treated with CCl_4_ showed hepatocyte degeneration and necrosis, the formation of fibrous tissue infiltrated with inflammatory cells, and distortion of the central venules. Masson’s staining (Appendix A) also demonstrated that the CCl_4_-induced rat model was successfully established.

### 3.2. Pharmacokinetic Study

The plasma concentration–time curves of EZE and its conjugated metabolites after the oral or intravenous administration of 10 mg/kg of EZE are shown in Figure 3 and Figure 4, respectively. The pharmacokinetic parameters are listed in Table 2. After intravenous injection, the EZE AUC_0–t_ value was close to that of EZE-Ph, which was approximately 2.5-fold higher than that of EZE-Hy. The AUC_0–t_ values for EZE, EZE-Ph, and EZE-Hy in CCl_4_-induced rats were 2.1-, 16.4-, and 4.3-fold higher than in control rats, respectively. Following oral administration, EZE-Ph was the main circulating component in plasma, with EZE and EZE-Hy accounting for only 0.4% and 11%, respectively. The AUC_0–t_ values for EZE, EZE-Ph, and EZE-Hy were 11.1-, 4.4-, and 2.5-fold higher in CCl_4_-induced rats than control rats.

To evaluate the role of the liver in the metabolism and the elimination of EZE and its conjugated metabolites, EZE was orally administered to rats at 10 mg/kg. Blood samples were simultaneously collected from hepatic portal and femoral veins at designated time points. Figure 5 shows plasma concentration–time curves for EZE and EZE-Ph. EZE-Hy was not evaluated due to its low plasma concentration. The AUC_0–t_ value of EZE in portal vein plasma (1853 ng·h/mL) was 33.0-fold higher than in femoral vein plasma (56.2 ng·h/mL) in control rats, whereas in CCl_4_-induced rats, it was only 1.84-fold higher in portal vein plasma (352 ng·h/mL) than femoral vein plasma (191 ng·h/mL), indicating a significantly decreased hepatic first-pass metabolism. The EZE-Ph AUC_0–t_ value in portal vein plasma (14,850 ng·h/mL) was 5.75-fold higher than in femoral vein plasma (2581 ng·h/mL) in control rats, whereas in CCl_4_-induced rats, it was only 1.28-fold higher in portal vein plasma (15,944 ng·h/mL) than femoral vein plasma (12,475 ng·h/mL), indicating a marked decrease in hepatic clearance. In the control rats, a pronounced second absorption peak was observed at 9.5 h after an oral administration (Figure 5A), indicating that EZE underwent a distinct enterohepatic circulation. By contrast, this second peak was markedly reduced in CCl_4_-induced rats, indicating reduced enterohepatic circulation. This also resulted in lower plasma exposure of EZE in the portal vein blood of CCl_4_-induced rats than control rats. It could be inferred that the biliary excretion of glucuronide metabolites also decreased.

### 3.3. Incubation of EZE with Rat Liver S9 Fractions and Inhibition of EZE Glucuronidation by Bile Acids

To investigate the effects of hepatic failure on the hepatic metabolism, EZE (0.3, 3, and 30 μM) was incubated with liver S9 fractions isolated from control and CCl_4_-induced rats. Compared with the control rats, the formation of EZE-Ph decreased by 8.80%, 12.7%, and 27.4% in the CCl_4_-induced rats (Figure 6B), whereas the residue of EZE increased by 4.15-, 5.00-, and 1.40-fold (Figure 6A), demonstrating that the activity of UGTs decreased following liver injury.

Consistent with previous studies [37,38,39], CA, GCA, and TCA increased abnormally in the serum of rats with hepatic failure (Appendix A). These three bile acids and their mixtures (rMix 1~3, Appendix A) were used to evaluate their effects on the glucuronidation of EZE. As shown in Appendix A, CA, GCA, TCA, and their mixtures, even up to 100 μM, exhibited only weak inhibition of EZE-Ph generation (<40%).

Recent studies have shown that CA, CDCA, UDCA, GCA, GCDCA, GUDCA, TCA, TCDCA, and TUDCA greatly increased in the serum of CLD patients [21,40]. Hence, their mixtures (hMix 1~3, Appendix A) were used to evaluate their effects on the glucuronidation of EZE. As shown in Appendix A, mixtures up to 100 μM exhibited weak inhibition of EZE-Ph generation (<40%), indicating that the effects of elevated bile acids on the activity of UGTs can be ignored in humans.

### 3.4. Rat Hepatocyte Uptake Studies and Inhibition of EZE-Ph Uptake by Bile Acids

The uptake of EZE into isolated rat hepatocytes was independent of temperature (Figure 7A), suggesting that EZE was absorbed via passive diffusion. The uptake of EZE-Ph at 37 °C was 50.0 times greater than that at 4 °C (Figure 7B), indicating that active transporters are involved in the EZE-Ph uptake into rat hepatocytes.

The inhibitory effects of the main rat bile acids on the uptake of EZE-Ph were evaluated (Figure 7C). The uptake by rat hepatocytes decreased to 12.0% in the presence of 200 µmol/L of rifampicin (positive inhibitor of OATPs). In the presence of 1, 10, and 100 μM CA, the EZE-Ph uptake decreased to 108%, 53.5%, and 20.1%, respectively. With 1, 10, and 100 μM GCA, the EZE-Ph uptake decreased to 109%, 64.1%, and 16.3%. With 1, 10, and 100 μM TCA, it decreased to 94.1%, 53.4%, and 13.9%. With 1, 10, and 100 μM bile acid mixtures (rMix 1~3), it decreased to 100%, 59.7%, and 7.63%, respectively. This showed that bile acids significantly inhibited the activity of hepatic uptake transporters.

### 3.5. Uptake of EZE-Ph in Liver Slices

The effect of hepatic impairment on the EZE-Ph uptake in the control and CCl_4_-induced liver slices was examined over 10 min. As shown in Figure 8, the EZE-Ph uptake was significantly greater at 37 °C than 4 °C, but no significant difference was observed between control and CCl_4_-induced rats, suggesting that the activity of hepatic uptake transporters was unaffected by liver injury.

### 3.6. Uptake of EZE-Ph into Transfected HEK293 Cells

To elucidate the roles of hepatic and renal uptake transporters in the disposition of EZE-Ph, its uptake by six transporters (OATP1B1, OATP1B3, OATP2B1, OAT1, OAT3, and OCT2) was examined. The function of these transporters has been previously validated using typical substrates (Appendix A). As shown in Appendix A, the uptake of 10 μM EZE-Ph by OATP1B1, OATP1B3, OATP2B1, and OAT3 were 5.3, 56, 3.7, and 8.9 times higher, respectively, in uptake transporter-transfected cells than in mock-transfected cells. In contrast, the uptake of EZE-Ph by OAT1- and OCT2-transfected cells was less than twice that of mock-transfected cells. This suggests that EZE-Ph is a substrate for OATP1B1, OATP1B3, OATP2B1, and OAT3, but not for OAT1 or OCT2.

The roles of OATP1B1, OATP1B3, OATP2B1, and OAT3 in the transport of EZE-Ph were further evaluated by measuring the time- and concentration-dependent uptake of EZE-Ph in transfected HEK293 cells. The rates of EZE-Ph uptake by transfected HEK293 cells increased in a linear time-dependent manner after incubation for 5 min (Appendix A). Thus, 5 min was chosen as the uptake time for the subsequent concentration-dependent assay. The OATP2B1-mediated EZE-Ph uptake exhibited a typical Michaelis–Menten curve (Figure 9C), whereas OATP1B1- and OATP1B3-mediated EZE-Ph uptake exhibited substrate inhibition kinetics (Figure 9A,B), and OAT3-mediated EZE-Ph uptake displayed a sigmoidal autoactivation profile (Figure 9D). The K_m_ and V_max_ values of EZE-Ph are shown in Table 3. OATP1B3 exhibited greater affinity for EZE-Ph than OATP1B1, OATP2B1, and OAT3 according to the estimated K_m_ value. The CL_int_ of EZE-Ph was highest in OATP1B3-transfected cells, showing that OATP1B3 is the major transporter involved in the hepatic uptake of EZE-Ph.

### 3.7. Inhibition of Bile Acids for the Uptake of EZE-Ph in OATP1B3-Transfected HEK293 Cells

The inhibitory effects of the main human bile acids and rifampicin on the uptake of EZE-Ph and estradiol-17β-glucuronide (positive substrate) were evaluated. As shown in Figure 10A,B, the uptake of EZE-Ph and estradiol-17β-glucuronide decreased to 7.94% and 3.89%, respectively, in the presence of 200 μM rifampicin. With 1, 10, and 100 μM bile acid mixtures (hMix 1~3), the EZE-Ph uptake decreased to 111%, 71.9%, and 14.6%, respectively, and estradiol-17β-glucuronide decreased to 95.9%, 67.8%, and 15.9%. This indicated that bile acids significantly inhibited the activity of the hepatic uptake transporter OATP1B3 and were comparable to the positive inhibitor rifampicin at high concentrations. Furthermore, the bile acid mixtures inhibited OATP1B3-mediated EZE-Ph and estradiol-17β-glucuronide uptake in a concentration-dependent manner, with IC_50_ values of 15.0 and 21.2 μM, respectively (Figure 10C,D), which are markedly lower than the levels of TBA (51.3 μM) observed in CLD patients [21].

The inhibitory effects of individual bile acids, which were significantly elevated in the serum of CLD patients, were investigated (Figure 11). GCDCA and TCDCA strongly inhibited the OATP1B3-mediated uptake of EZE-Ph, with IC_50_ values of 15.1 and 7.94 μM, respectively, which are lower than their levels in CLD patients (17.3 and 9.18 μM) [21]. Since the IC_50_ values for CA, GCA, TCA, CDCA, UDCA, GUDCA, and TUDCA were higher than their concentrations in CLD patients, their inhibitory effects can be ignored.

### 3.8. Efflux Transport Studies with Membrane Vesicles

To elucidate the role of hepatic efflux transporters in the disposition of EZE and EZE-Ph, we examined their efflux by three transporters, namely MDR1, MRP2, and MRP3. No significant ATP-dependent transport of EZE was observed, suggesting that the parent drug was not a substrate for these transporters (Figure 12A). The ratios of ATP-dependent uptake to nonspecific adsorption (+ATP/-ATP ratio) of EZE-Ph via MDR1-, MRP2-, and MRP3-expressing membrane vesicles were 5.64, 65.5, and 5.71 (Figure 11B), respectively, demonstrating that EZE-Ph was a substrate for all three transporters.

The inhibitory effects of the main human bile acids on the efflux of EZE-Ph were also evaluated. As shown in Figure 12B, the 100 μM bile acid mixture (hMix 3) reduced the EZE-Ph uptake by MDR1, MRP2, and MRP3 to 51.7%, 107%, and 65.4%, respectively, indicating that the bile acids did not inhibit, or only weakly inhibited, the efflux of EZE-Ph (with IC_50_ values above 100 μM).

### 3.9. Intestinal and Hepatic Gene Expression of Efflux Transporters in Rats

EZE was extensively metabolized to its glucuronide in the intestine. Hardwick et al. evaluated changes in the hepatic gene expression of efflux transporters in NASH rats [33]. The present study determined both the hepatic and intestinal mRNA expression of *Mdr1a*, *Mdr1b*, *Mrp2*, and *Mrp3* in control and CCl_4_-induced rats. *Mdr1a* and *Mdr1b* mRNA levels were significantly increased (2- to 3-fold) in the various segments of the intestine of CCl_4_-induced rats (Figure 13A,D), but *Mrp2* was significantly decreased (60–80%), except in the colon. *Mrp3* was not altered at the transcriptional level. In CCl_4_-induced rat livers (Figure 13E), the *Mdr1b* and *Mrp3* mRNA levels were greatly elevated (192- and 9.51-fold, respectively), which is consistent with a previous finding [33]. However, *Mrp2* significantly decreased (56%). No significant changes were observed in the *Mdr1a* mRNA levels.

## 4. Discussion

The plasma exposure of EZE and its glucuronide conjugates markedly increased in CLD patients compared to healthy subjects. In this study, a CCl_4_-induced hepatic failure rat model was used to investigate the effects of hepatic impairment on the pharmacokinetics of EZE and EZE-Ph. It has been documented that following the intraduodenal administration of EZE, EZE-Ph is rapidly generated (>95%) in the intestine and excreted through bile [28]. The current pharmacokinetic study in the portal and femoral vein blood of control and CCl_4_-induced rats also confirmed that EZE experienced a strong intestinal first-pass metabolism. The AUC_0-t_ value of EZE-Ph in portal vein blood was similar in control and CCl_4_-induced rats, whereas it increased 4.83-fold in the femoral vein blood of treated rats, suggesting that hepatic insufficiency did not affect the intestinal first-pass metabolism but significantly reduced hepatobiliary elimination. The reduced biliary excretion of EZE-Ph led to decreased enterohepatic circulation of EZE; thus, the AUC_0-t_ of portal vein EZE in CCl_4_-induced rats decreased by 81.0% compared with control rats. The ratio of the AUC_0–t_ of EZE in portal vein and femoral vein blood was 33.0 in the control rats, suggesting that the EZE that was not metabolized in the intestine continued to be extensively metabolized in the liver. By comparison, this ratio was only 1.84 in CCl_4_-induced rats, indicating that liver injury reduced the hepatic metabolism of EZE, which might be the cause of the increased plasma exposure of EZE. Since only a small proportion of EZE-Ph was generated in the liver after oral administration, the EZE-Ph production was not affected by liver injury, but its hepatobiliary elimination was significantly reduced, which could result in increased plasma exposure.

In vitro studies of EZE and EZE-Ph were conducted to confirm this hypothesis. The data revealed that EZE was not a substrate for uptake or efflux transporters; hence, the influence of hepatic failure on the glucuronidation of EZE was evaluated. The metabolism of EZE to its phenolic glucuronide in the liver decreased compared with control rats, and the lower the EZE concentration, the greater the impact of liver injury, which explained the finding that EZE exposure was markedly increased only by oral, but not intravenous, administration in CCl_4_-induced rats. Recent in vivo studies have shown that liver disease can result in the accumulation of bile acids, which may have inhibitory effects on UGTs. For example, lithocholic acid strongly inhibited the glucuronidation of 4-methylumbelliferone [41]. Therefore, the inhibitory effects of bile acids on the hepatic metabolism of EZE were also examined, but these could be ignored since their IC_50_ values (>100 μM) were far above the serum concentrations observed in patients (Appendix A). Thus, the decreased activity of UGTs in the liver led to the elevated plasma exposure of EZE after intravenous or oral administration.

The hepatic uptake of multiple endobiotics and xenobiotics from the sinusoidal blood depends on specific protein carriers, such as OATPs and NTCP. EZE-Ph undergoes extensive enterohepatic recycling; thus, hepatic uptake transporters may be rate-limiting determinants. Rat hepatocyte uptake studies showed that EZE-Ph may be a substrate of OATP(s). Interspecies differences for the function and expression of OATPs between animals and humans have been reported [42]. Thus, the uptake of EZE-Ph into transfected HEK293 cells was further examined, revealing that EZE-Ph was a substrate for OATP1B1, OATP1B3, and OATP2B1 (Appendix A), with OATP1B3 dominating (CL_int_ was 6.73 μL/min/mg protein). Different from the reported results [30], EZE-Ph was not only a substrate for OATP1B1 and OATP2B1, but also for OATP1B3, and its uptake increased in a time- and concentration-dependent manner. The values of kinetic parameters, including K_m_, V_max_, and CL_int_, were also firstly shown. Although there was no significant difference in the uptake of EZE-Ph between the control and CCl_4_-induced livers, bile acid mixtures inhibited the uptake of EZE-Ph into rat hepatocytes. Recent studies have shown that the concentrations of TBA were 10.7-fold higher in CLD patients (51.3 μM) than in healthy subjects. Among the individual bile acids, GCA, TCA, GCDCA, and TCDCA were 28.7-, 97.2-, 13.8-, and 53.3-fold higher, respectively, in CLD patients than healthy subjects [21]. This study evaluated the inhibitory effects of individual and mixtures of bile acids on the OATP1B3-mediated uptake of EZE-Ph. Bile acid mixtures significantly inhibited EZE-Ph uptake (IC_50_ 15.0 μM), while the GCDCA and TCDCA IC_50_ values (15.1 and 7.94 μM, respectively) were markedly below the concentrations observed in CLD patients. Thus, the hepatic extraction of EZE-Ph in the basolateral membrane significantly decreased, leading to elevated systemic exposure. Consistent with previous studies [31,43], EZE-Ph is a substrate for MDR1, MRP2, and MRP3. Thus, the mRNA expression of *Mdr1a*, *Mdr1b*, *Mrp2*, and *Mrp3* was measured in the duodenum, jejunum, ileum, colon, and liver. The *Mdr1a* and *Mdr1b* mRNA levels were significantly increased (2- to 3-fold) in the duodenum, jejunum, and ileum of CCl_4_-induced rats, whereas the *Mrp2* mRNA significantly decreased (60–80%). The regulatory effects of efflux transporters were contradictory, but the pharmacokinetic changes in EZE-Ph suggested that liver injury reduced the intestinal efflux. This was because the concentration of the first absorption peak of EZE-Ph in the portal vein of CCl_4_-induced rats significantly increased after oral administration. In CCl_4_-induced rat livers, the *Mdr1b* and *Mrp3* mRNA levels were both greatly elevated (192-fold and 9.51-fold, respectively), which is consistent with a previous study [33]. However, the *Mrp2* mRNA levels decreased significantly (56%). Although the expression of *Mdr1* was significantly elevated, Hardwick et al. proved that its localization changed in NASH rats, disrupting biliary drug efflux [33]. Thus, the combination of the induced *Mrp3* expression, the decreased *Mrp2* expression, and the altered localization of *Mdr1* in CCl_4_-induced rats shifted the disposition profile of glucuronide conjugates toward plasma retention.

This study found that changes in elevated exposures of EZE and EZE-Ph were different following oral administration and intravenous injection, which might be related to the site of the first-pass metabolism of EZE and the localization of the transport of EZE-Ph. In contrast to intravenous administration, EZE was mainly metabolized to its conjugated metabolites in the intestine, and only a very small amount in the liver, after oral administration. The in vitro liver S9 experiment suggested that hepatic injury decreased the hepatic UGT activity in a concentration-dependent manner. The lower the EZE concentration, the greater the impact of liver injury, which might explain the finding that ezetimibe exposure was markedly increased only by oral, but not intravenous, administration in CCl_4_-treated animals. In the present study, it was observed that there was no significant difference in the portal blood exposure to EZE-Ph between the control and CCl_4_-induced rats, indicating that the activity of intestinal UGTs was not affected by liver injury. EZE-Ph was generated in the enterocytes after oral administration, then transported into the liver via OATPs, and excreted into bile via MRP2 and MDR1. Thus, its exposure is mainly influenced by uptake transporters. However, after intravenous administration, EZE-Ph was generated in the hepatocytes, then excreted into bile via MRP2 and MDR1 or into the sinusoidal blood via MRP3, and subsequently transported into the liver again via OATPs. Thus, its plasma exposure is influenced by efflux and uptake transporters. Consequently, the change in the plasma exposure of EZE-Ph was greater after intravenous injection than after oral administration. In addition, as a substrate of OAT3, EZE-Ph was excreted in urine to compensate for impaired hepatobiliary elimination during liver disease.

The mechanisms of changes in the disposition of EZE and its glucuronides caused by hepatic dysfunction are shown in Figure 14. Essentially, decreased activity of UGTs in the liver (resulting in a reduced hepatic first-pass metabolism) led to the elevated plasma exposure of EZE. The combination of the induced expression of *Mrp3*, the decreased expression of *Mrp2*, the altered localization of *Mdr1*, and the inhibition of OATPs by bile acids resulted in the elevated plasma exposure of EZE-Ph.

In conclusion, the pharmacokinetics of circulating substances undergoing hepatic metabolism or elimination may be altered under liver disease conditions, and the degree of change can be influenced by the dosing routes. In addition to changes in the hepatic metabolizing enzymes and transporters themselves, the inhibitory effects of accumulated endogenous compounds on their activities should also be considered.

## Figures and Tables

**Figure 1 pharmaceutics-14-02743-f001:**
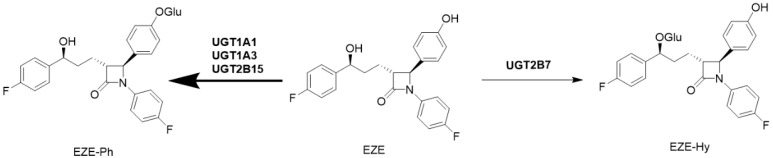
Major metabolic pathways of EZE in humans.

**Figure 2 pharmaceutics-14-02743-f002:**
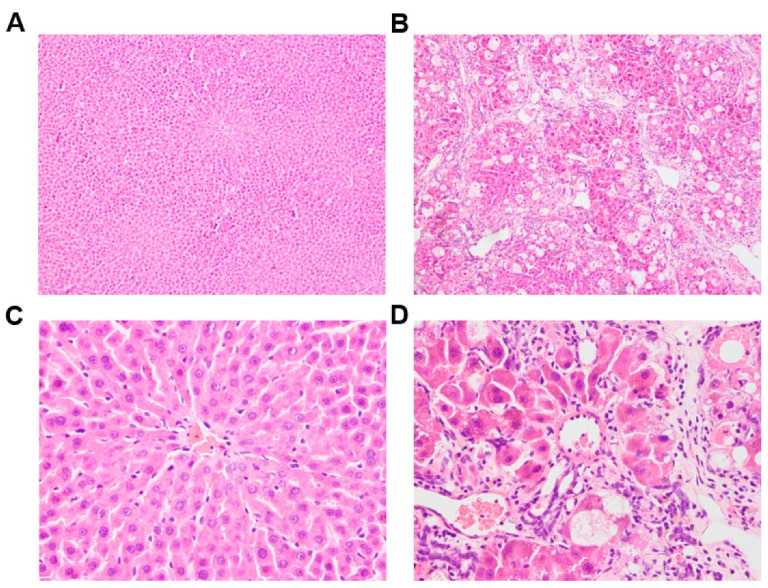
Histology of liver tissue from control ((**A**): 100×, (**C**): 400×) and CCl_4_-induced ((**B**): 100×, (**D**): 400×) rats using hematoxylin–eosin staining.

**Figure 3 pharmaceutics-14-02743-f003:**
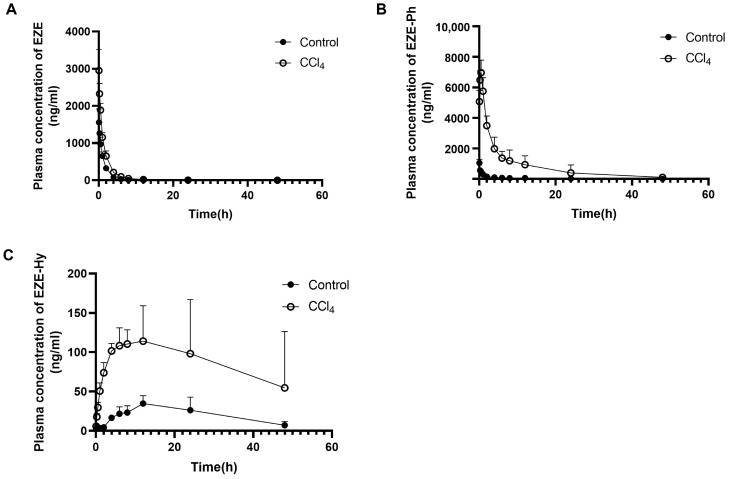
Mean plasma concentration–time profiles of EZE (**A**), EZE-Ph (**B**), and EZE-Hy (**C**) following intravenous administration of 10 mg/kg of EZE to control and CCl_4_-induced rats. Each point was presented as mean ± S.D. (*n* = 5).

**Figure 4 pharmaceutics-14-02743-f004:**
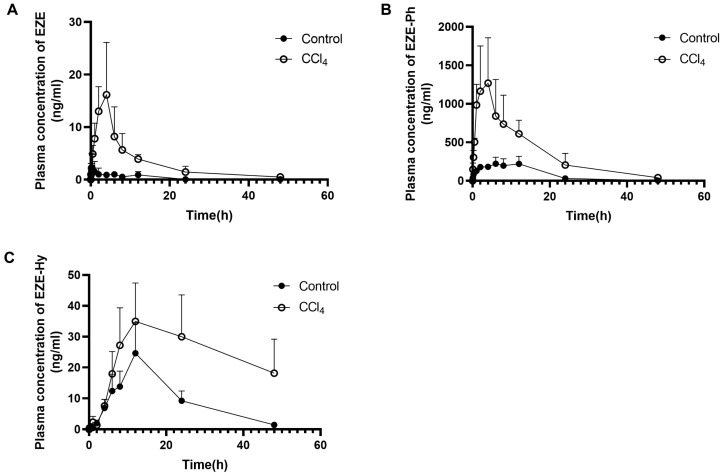
Mean plasma concentration–time profiles of EZE (**A**), EZE-Ph (**B**), and EZE-Hy (**C**) following oral administration of 10 mg/kg EZE to control and CCl_4_-induced rats. Each point was presented as mean ± S.D. (*n* = 5).

**Figure 5 pharmaceutics-14-02743-f005:**
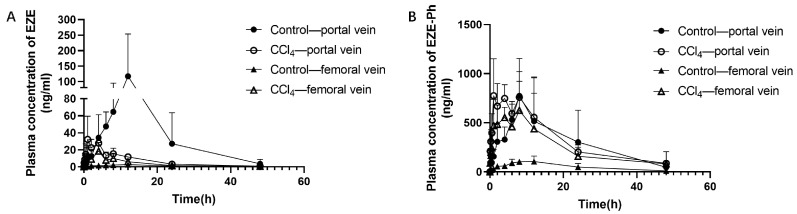
Mean plasma concentration–time profiles of EZE (**A**) and EZE-Ph (**B**) in portal vein and femoral vein blood following oral administration of 10 mg/kg of EZE to control and CCl_4_-induced rats. Each point was presented as mean ± S.D. (*n* = 5).

**Figure 6 pharmaceutics-14-02743-f006:**
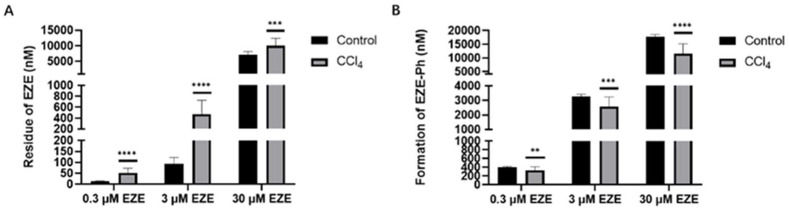
Residue of EZE (**A**) and formation of EZE-Ph (**B**) in liver S9 of control and CCl_4_-induced rats after incubation with 0.3 μM, 3 μM, and 30 μM EZE for 60 min. Data are expressed as mean ± S.D. (*n* = 3). ** *p* < 0.01, *** *p* < 0.001, and **** *p* < 0.0001 compared with control.

**Figure 7 pharmaceutics-14-02743-f007:**
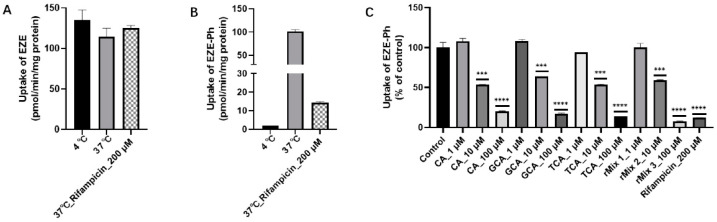
Uptake of EZE (**A**) and EZE-Ph (**B**), and inhibitory effects of bile acids and rifampicin on EZE-Ph uptake (**C**) by rat primary hepatocytes over 10 min. Data are expressed as mean ± S.D. (*n* = 3). *** *p* < 0.001 and **** *p* < 0.0001 compared with control.

**Figure 8 pharmaceutics-14-02743-f008:**
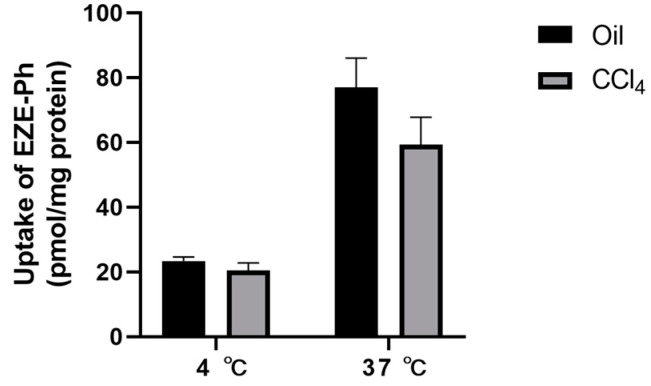
Uptake of EZE-Ph in liver slices of control and CCl_4_-induced rats. Data are expressed as mean ± S.D. (*n* = 3).

**Figure 9 pharmaceutics-14-02743-f009:**
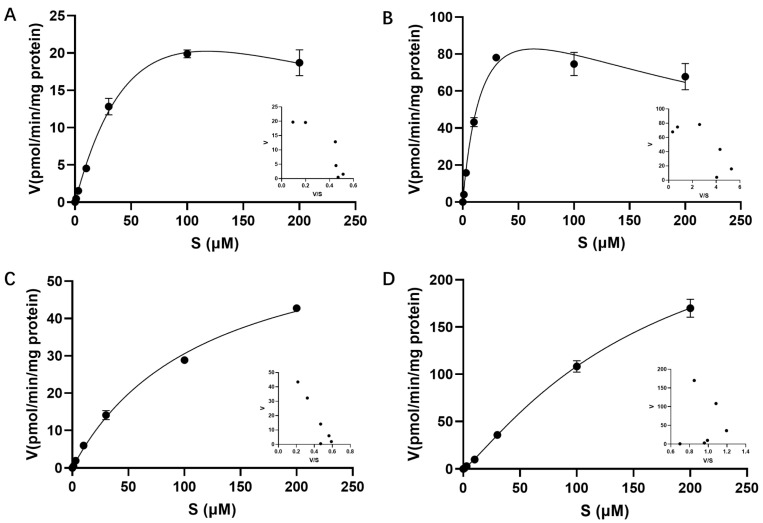
Concentration-dependent uptake of EZE-Ph into OATP1B1- (**A**), OATP1B3- (**B**), OATP2B1- (**C**), and OAT3- (**D**) expressing HEK293 cells. The uptake time was 5 min. Each point was presented as mean ± S.D. (*n* = 3).

**Figure 10 pharmaceutics-14-02743-f010:**
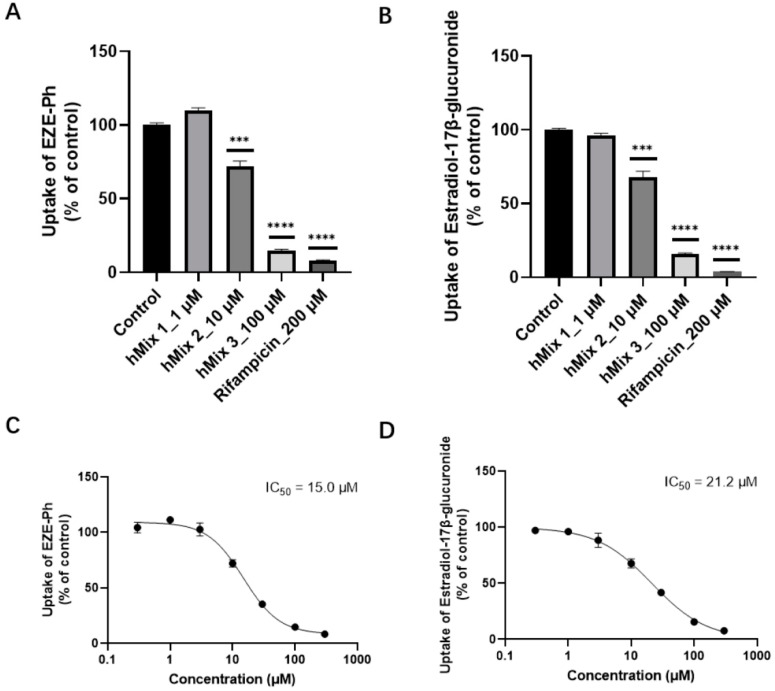
Inhibitory effects of bile acid mixtures and rifampicin on EZE-Ph (**A**) and estradiol-17β-glucuronide (**B**) uptake in OATP1B3-expressing HEK293 cells, and OATP1B3-mediated uptake of 10 μM EZE-Ph over 5 min (**C**) and uptake of 5 μM estradiol-17β-glucuronide over 4 min (**D**) in the absence and presence of bile acid mixtures. OATP1B3-mediated EZE-Ph and estradiol-17β-glucuronide accumulations were corrected by subtracting the nonspecific accumulation in mock-transfected HEK293 cells from that in OATP1B3-expressing HEK293 cells. The values were expressed as a percentage of the uptake in the absence of bile acid mixtures. Solid lines represent the fitted line obtained by nonlinear regression analysis. Data are expressed as mean ± S.D. (*n* = 3). *** *p* < 0.001 and **** *p* < 0.0001 compared with control.

**Figure 11 pharmaceutics-14-02743-f011:**
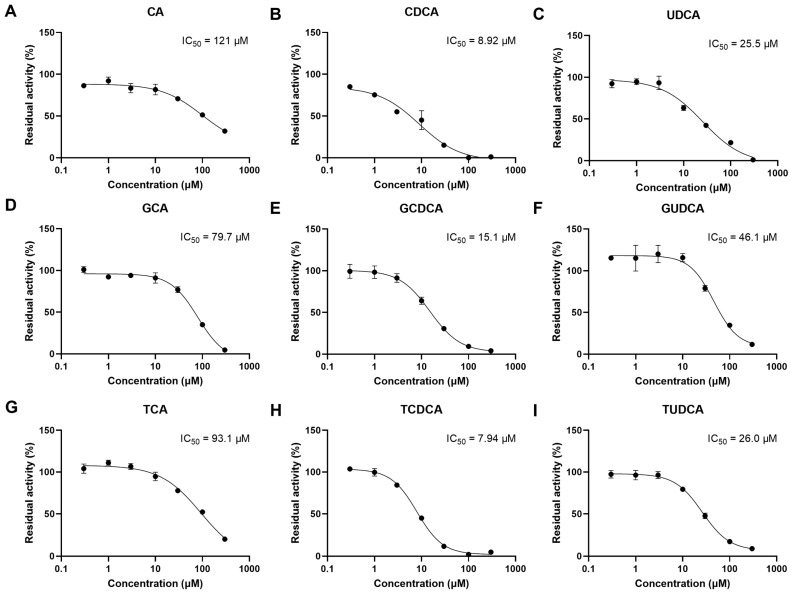
OATP1B3-mediated uptake of 10 μM EZE-Ph over 5 min in the absence and presence of CA (**A**), CDCA (**B**), UDCA (**C**), GCA (**D**), GCDCA (**E**), GUDCA (**F**), TCA (**G**), TCDCA (**H**), and TUDCA (**I**). OATP1B3-mediated EZE-Ph accumulations were corrected by subtracting the nonspecific accumulation in mock-transfected HEK293 cells from that in OATP1B3-expressing HEK293 cells. The values were expressed as a percentage of the uptake in the absence of bile acids. Solid lines represent the fitted line obtained by nonlinear regression analysis. Each point was presented as mean ± S.D. (*n* = 3).

**Figure 12 pharmaceutics-14-02743-f012:**
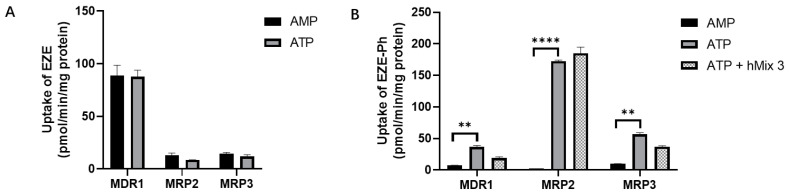
(**A**) Direct transport of 10 μM EZE via MDR1, MRP2, and MRP3, and (**B**) inhibitory effects of bile acid mixtures on the direct transport of 10 μM EZE-Ph by MDR1, MRP2, and MRP3. Compounds were incubated for 10 min with transporter-expressing membrane vesicles with or without ATP. Data were expressed as mean ± S.D. (*n* = 3). ** *p* < 0.01 and **** *p* < 0.0001 compared with control.

**Figure 13 pharmaceutics-14-02743-f013:**
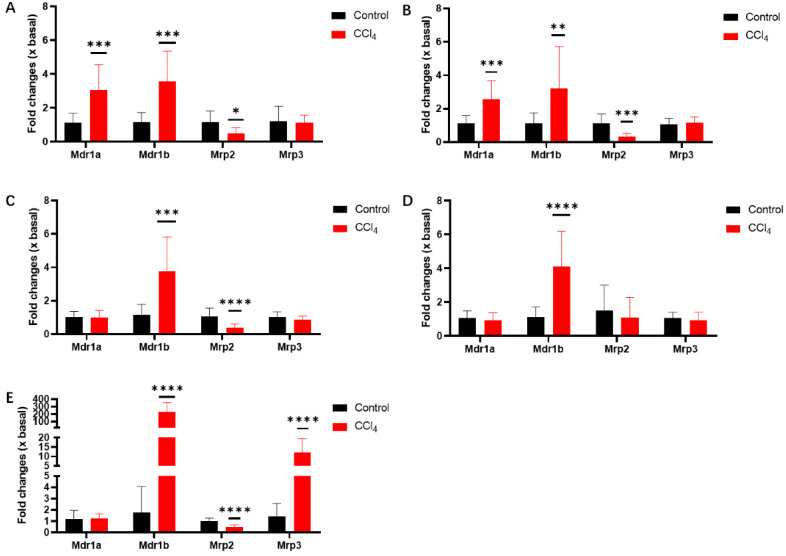
mRNA expression of *Mdr1a*, *Mdr1b*, *Mrp2*, *and Mrp3* in duodenum (**A**), jejunum (**B**), ileum (**C**), colon (**D**), and liver (**E**) of control and CCl_4_-induced rats. Data were expressed as mean ± S.D. (*n* = 12). * *p* < 0.05, ** *p* < 0.01, *** *p* < 0.001, and **** *p* < 0.0001 compared with control.

**Figure 14 pharmaceutics-14-02743-f014:**
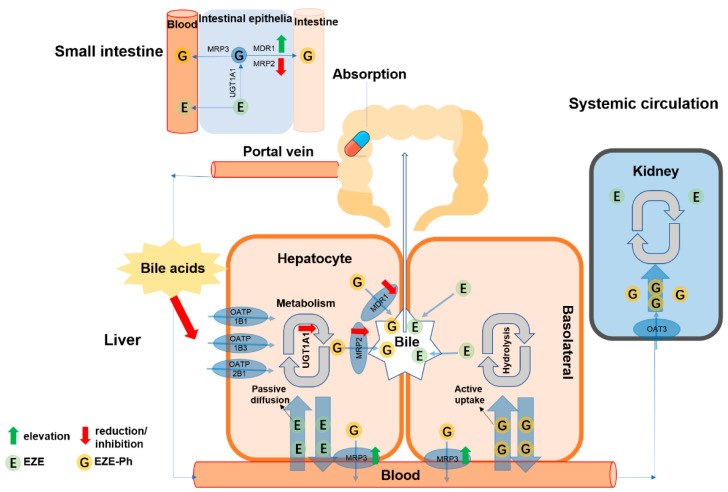
The in vivo pharmacokinetics of EZE, its major metabolites, and the proposed mechanism of the effects of hepatic impairment on these circulating compounds.

**Table 1 pharmaceutics-14-02743-t001:** The liver index and biochemical parameters of control and CCl_4_-induced rats. Data are expressed as mean ± S.D. (*n* = 16).

Parameter	Control Rats	CCl_4_-Induced Rats
Liver weight/body weight	31.1 ± 3.4	37.9 ± 5.9 ^c^
ALT (U/L)	13.0 ± 5.4	112 ± 33 ^d^
AST (U/L)	21.7 ± 6.2	125 ± 34 ^d^
AKP (U/L)	2562 ± 674	9887 ± 1633 ^d^

^c^*p* < 0.001 compared with control, ^d^
*p* < 0.0001 compared with control.

**Table 2 pharmaceutics-14-02743-t002:** Pharmacokinetic parameters of EZE and its major metabolites after intravenous and oral administration of 10 mg/kg of EZE to control (*n* = 5) and CCl_4_-induced rats (*n* = 5). Data are expressed as mean ± S.D.

Route of Administration	Group	Pharmacokinetic Parameters	EZE	EZE-Ph	EZE-Hy
i.v.	Control rats	C_max_ (ng/mL)	1558 ± 361 *	1047 ± 240	36.9 ± 9.6
		t_max_ (h)	-	0.083 ± 0.00	16.8 ± 6.6
		t_1/2_ (h)	6.76 ± 3.31	8.96 ± 2.19	16.1 ± 7.6
		AUC_0–t_ (ng·h/mL)	2181 ± 304	2467 ± 418	992 ± 392
	CCl_4_-induced rats	C_max_ (ng/mL)	2952 ± 571 *^,b^	6972 ± 807 ^d^	146 ± 42 ^c^
		t_max_ (h)	-	0.45 ± 0.11 ^b^	11.2 ± 7.6
		t_1/2_ (h)	7.67 ± 3.08	9.58 ± 6.20	15.3 ± 2.4
		AUC_0–t_ (ng·h/mL)	4612 ± 781 ^c^	40450 ± 1810 ^b^	4247 ± 2051 ^b^
p.o.	Control rats	C_max_ (ng/mL)	2.63 ± 2.68	276 ± 69	24.6 ± 9.9
		t_max_ (h)	4.95 ± 4.84	8.80 ± 4.60	12.0 ± 0.0
		t_1/2_ (h)	5.56 ± 3.78	5.50 ± 0.36	-
		AUC_0–t_ (ng·h/mL)	17.0 ± 10.0	4174 ± 1140	463 ± 160
	CCl_4_-induced rats	C_max_ (ng/mL)	20.4 ± 8.1 ^b^	1342 ± 585 ^b^	38.1 ± 10.1
		t_max_ (h)	5.20 ± 3.90	3.00 ± 1.41	14.4 ± 5.4
		t_1/2_ (h)	12.3 ± 5.3	8.98 ± 2.10	-
		AUC_0–t_ (ng·h/mL)	189 ± 66 ^c^	18255 ± 6993 ^b^	1172 ± 418 ^b^

* represents the plasma concentration at 5 min after intravenous injection. AUC_0–t_, area under the concentration–time curve from 0 h to the last sampling time; C_max_, maximum plasma concentration; t_max_, time to the C_max_; t_1/2_, apparent elimination half-life. ^b^
*p* < 0.01 compared with control, ^c^
*p* < 0.001 compared with control, ^d^
*p* < 0.0001 compared with control.

**Table 3 pharmaceutics-14-02743-t003:** Kinetic parameters for the uptake of EZE-Ph into human OATP1B1-, OATP1B3-, OATP2B1-, and OAT3-expressing HEK293 cells.

Transporter	Kinetic Model	K_m_ (μM)	V_max_ (pmol/min/mg Protein)	CL_int_ (μL/min/mg Protein)	h
OATP1B1	Substrate inhibition	91.5	51.7	0.56	
OATP1B3	Substrate inhibition	19.9	134	6.73	
OATP2B1	Michaelis–Menten	119	66.9	0.56	
OAT3	Allosteric sigmoidal	166	306	1.84	1.19

## Data Availability

The data presented in this study are available in this article and Appendix A.

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
