# Peer review of "Changes in Disposition of Ezetimibe and Its Active Metabolites Induced by Impaired Hepatic Function: The Influence of Enzyme and Transporter Activities"

_pharmaceutics, 2022, doi:10.3390/pharmaceutics14122743_

Round 1

Reviewer 1 Report (Previous Reviewer 1)

Authors have addressed my comments except that related to the conclusion section in the abstract. This part must be concise and short. Authors are kindly requested to rewrite this part in a way that describe their most important finding and recommendation in only two-three sentences. 

Author Response

Reviewer 2 Report (Previous Reviewer 2)

The authors have considered most of my comments in a satisfactory manner. However, some changes are still required:

1. As already suggested in my initial evaluation, the authors should add to the title of the manuscript that the study was performed in rats (i.e. "Changes in Disposition of Ezetimibe and its Metabolites by Chemically Induced Hepatic Impairment in Rats: The Influence of Enzyme and Transporter Activities". Otherwise, the title is misleading as the reader will assume a study in humans.

2. The data shown in Figure 9 and Table 3 should be discussed to comparable data already published in 2008 (Reference 31).

3. I am not satisfied with authors response to my former comments 11 and 12. In general, I agree that measuring function may be an excuse for not analyzing protein data. However, the PK of EZE is very complex. Thus, any changes can be attributed to different changes of enzymes or transporters. For example, increased serum levels of EZE Gluc may be due to a) decrease in hepatic OATPs, b) increase of hepatic and intestinal UGTs, c) decrease of intestinal and hepatic MRP2 / P-gp or, d). increase of basolateral MRP3 in intestine and liver. Thus, it is nearly impossible to jump to any conclusion form activity data alone. Thus, the authors should provide expression data for intestinal and hepatic tissue for the relevant determinants (P-gp, Mrp2/3, Bcrp, Oatps, Ugts).

Author Response

Reviewer 3 Report (New Reviewer)

Journal: Pharmaceutics (ISSN 1999-4923)

Manuscript ID: pharmaceutics-2049808

Changes in Disposition of Ezetimibe and its Active Metabolites Induced by Impaired Hepatic Function: The Influence of Enzyme and Transporter Activities

Ningjie Xie , Hong Wang , Hua Qin , Zitao Guo , Hao Xue , Jiafeng Hu , Xiaoyan Chen.

Abstract: Ezetimibe (EZE) is a selective cholesterol absorption inhibitor. Hepatic impairment significantly increased systemic exposure of EZE, and its main active phenolic glucuronide, EZE-Ph. Although changes in efflux transporter activity partly explain changes in EZE-Ph pharmacokinetics, the causes of changes to EZE and the effects of administration route on EZE-Ph remain unclear. A carbon tetrachloride (CCl4)-induced hepatic failure rat model was combined with in vitro experiments to explore altered EZE and EZE-Ph disposition caused by hepatic impairment. Plasma exposure of EZE and EZE-Ph increased 11.1- and 4.4-fold in CCl4-induced rats following oral administration of 10 mg/kg EZE, and 2.1- and 16.4-fold after intravenous injection. Conversion of EZE to EZE-Ph decreased concentration-dependently in CCl4-induced rat liver S9 fractions, but no change was observed in intestinal metabolism. EZE-Ph was a substrate for multiple efflux and uptake transporters, unlike EZE. In contrast to efflux transporters, no difference was seen in hepatic uptake of EZE-Ph between control and CCl4-induced rats. However, bile acids accumulating due to liver injury inhibited the uptake of EZE-Ph by organic anion transporting polypeptides (OATPs) (glycochenodeoxycholic acid and taurochenodeoxycholic acid had IC50 values of 15.1 and 7.94 μM in OATP1B3-overexpressed cells). In conclusion, increased plasma exposure of the parent drug EZE during hepatic dysfunction was attributed to decreased hepatic glucuronide conjugation, whereas the increased exposure of the metabolite EZE-Ph was mainly related to transporter activity. After oral administration, the inhibitory effects of bile acids on OATPs elevated plasma EZE-Ph. However, after intravenous administration EZE-Ph was more affected by liver injury, due to the combined effects of uptake and efflux transporters. It will help patients with hepatic impairment to take EZE safely and to predict possible enzyme and transporter based drug-drug interactions.

Comments: It is a topic of great interest to researchers in the related area (CLD).
It also provides relevant information on possible enzyme- and transporter-based drug-drug interactions.
In my opinion, I consider that the paper needs minor improvements before being accepted for publication.
My detailed comments are as follows:
  I have not detected any excessive citations of the authors in the manuscript. It seems verified that the authors
work in the field of research. The materials and methods of the manuscript are correctly described (an extensive
methodological battery is included, which clearly supports the results and conclusions provided), especially the
part that corresponds to pharmacokinetic experiments. The results of HE are clear (histopathological changes
are characteristic of the model). The results showed a very important structural disorder.
In my opinion, I would add other stains that give more quality to the work (sirius red, Masson, f4/80......

I have a question, which will surely be easy to answer (The manuscript shows that the animal study has been
approved by the authorities and that is beyond any doubt). Blood samples were collected from the retroorbital
venous plexus predose (0 hours), at 5, 15, and 30 minutes, and at 1, 2, 4, 6, 8, 12, and 24 hours postdose into
tubes containing 2K-EDTA. The RO extraction in the same animal and during all the times that appear in the
manuscript?. With anesthetized animals?, there are 10 extractions (RO) in the same animal....

Another important aspect that would give more visibility to the results is the format of some of the figures in the
manuscript. As a suggestion, I would increase the sizes of figures 4, 5 and 11.

I believe, once again as I mentioned before, that minimal interventions and contributions are necessary to  increase the (already good) quality of the work.

Round 2

Reviewer 2 Report (Previous Reviewer 2)

The authors responded to all of my additional questions in a satisfactory manner. Its up to decide by the Editor, whether my suggestion to clarify in the title that this study was performed in rats has to be realized by the authors. I do not understand why the authors are not willing to show this information in the title and will not invest more time to this discussion.

This manuscript is a resubmission of an earlier submission. The following is a list of the peer review reports and author responses from that submission.

Round 1

Reviewer 1 Report

1. No conclusion section is mentioned in the abstract. Authors must write 2-3 sentences at the end of the abstract about their finding.

2. Ezetimibe is a water insoluble compound, how it was administered intravenously??

3. Line 166; what is meant by “was spun at 9000 g”?? Please clarify and use scientific expression.

4. Many experiments are vague and authors did not specify how they were conducted the practical work such as details for histopathological study and extraction of RNA from the intestine and liver samples.

5. Tables should stand alone. All abbreviations should be mentioned as footnote.

6. Authors must refer to the supplementary data during the results and discussion as there are many figures that have been mentioned in the manuscript but they are submitted as supplementary data such as S1A, S1B. …..etc.

7. One important thing about the pharmacokinetics of the parent compound and its metabolites;

·         Lines 74-76: Authors have mentioned that EZE undergoes intense intestinal first-pass metabolism to form predominantly a phenolic glucuronide and to a lesser extent a benzylic glucuronide.

·         Lines 496-497: EZE experienced strong intestinal first-pass metabolism.

Interestingly, data for the IV pharmacokinetics (table 2) showed an AUC of 2181 ± 304, 2467 ± 418 and 992 ± 392 for EZE, EZE-Ph and EZE-Hy, respectively in control rats and 4612 ± 781, 40450 ± 18101 and 4247 ± 2051, respectively in CCl4-induced rats.

It is not expected for a drug that has been administered by IV to undergo intestinal first-pass metabolism and so the data presented for AUC of EZE-Ph and EZE-Hy after IV administration is not correct otherwise you give a reasonable justification.  

Reviewer 2 Report

Comments on the manuscript entitled „Changes in Disposition of Ezetimibe and its Metabolites Induced by Impaired Hepatic Function: The Influence of Enzyme 3 and Transporter Activities”

Xie et al. investigated in their study the impact of chemically induced hepatic impairment on the disposition of the cholesterol absorption inhibitor ezetimibe in rats. As ezetimibe is extensively metabolized by UGT1A1/3 and the parent as well as the glucuronide metabolite are considered by several transporters such as P-gp, BCRP, MRP2, MRP3 and OATP1B1/3, remarkable changes could be anticipated. Indeed, the authors could show that CCl4-treatment resulted in markedly increased serum exposure of ezetimibe (EZE) and the phenolic glucuronide (EZE-Gluc). Interestingly, this effect was more pronounced for the parent compound after oral administration and for EZE-Gluc after intravenous administration. The authors argue that the observed changes are most like caused by the accumulation of bile acids in CCl4-treated animals. However, they have not measured the bile acids in hepatic tissue or plasma. Moreover, potential changes in the expression of hepatic uptake transporters, which is well established in liver disease, has not been investigated.  

In conclusion, there are from my perspective several serious uncertainties and methodical issues of the study, which need further clarification and additional experiments. Thus, I cannot suggest acceptance of the manuscript.  

I have the following comments / questions:

1.       The authors should add the important information to the manuscript title that the study was performed in rats and that hepatic impairment was chemically induced. I suggest the following title: “Changes in Disposition of Ezetimibe and its Metabolites by Chemically Induced Hepatic Impairment in Rats: The Influence of Enzyme and Transporter Activities”.

2.       The use of carbon tetrachloride (CCl4) to induced hepatic failure is a very drastic approach. The authors need to convince the readers that this experimental model is comparable to clinical settings of CLD such hepatitis B/C, nonalcoholic fatty liver disease, alcohol-associated liver disease, liver cirrhosis or hepatocellular carcinoma. For this purpose, the authors should extensively discuss the regulations of enzymes and transporters by the respective diseases and CCl4-treatment.

3.       The Introduction in its present form lacks any information on the involvement of drug transporters such as P-gp, BCRP, MRP2, MRP3 and OATP1B1/3 in the pharmacokinetics of ezetimibe although this is well established for years. This information is an essential prerequisite for the study hypothesis and should be added.

4.       Likewise, the changes in hepatic transporter expression by CLD should be added.

5.       Minor experimental comment: Please give centrifugation force in “g” and not “rpm” as this is a device-specific value.

6.       What was the reason for not measuring tissue concentrations of EZE and its metabolite in liver and intestinal sections? This information would be of great interest to prove the conclusions of the study (i.e. diminished hepatic uptake). If possible, these data should be generated.

7.       The experimental section on in vitro studies with Sf9 fractions, hepatocytes and transfected HEK293 cells lacks detailed information on the incubation procedure (time, concentration) with substrates (EZE / EZE-Gluc) and inhibitors. This should be added.

8.       The authors should provide convincing information that their used  OATP1B1-, OATP1B3-, OATP2B1-, OAT1-, OAT3- and OCT2-transfected HEK293 cells are working appropriately using established probe substrates. This is very important as there are countless clones circulating which show questionable functionality. This information should be provided as Supplemental data.

9.       It is a bit confusing that the authors used rats as in vivo model but performed in vitro-studies with human transport proteins. Considering that these SLC transporters show only a very limited amino acid homology of about 75% (i.e. they are ortholog proteins), this is a bit like comparing apples with oranges. This should be discussed in the manuscript. The same is true for studying human ABC transporters although their protein homology is much better than that of SLC transporters between human and rats.

10.    It is unclear, why the authors performed studies to show affinity of EZE / EZE-Gluc to human drug transporters. There are already many studies from different groups available showing this (no novelty). Please explain.

11.    What was the reason for not analyzing the gene expression of relevant hepatic Oatp transporters in rat liver. So far, only Mdr1a/b, Mrp2 and Mrp3 have been analyzed. This is not sufficient! Likewise, Ugt1a1 should be measured to exclude impact on glucuronidation by CCl4. Also intestinal tissue should be considered as there might be also involvement of changes in intestinal metabolism / transport.

12.    Nowadays, gene expression data alone are no more acceptable as it is known that there is often (especially in case of diseases) a disconnection of gene expression and protein abundance. The authors are encouraged to provide protein data for Ugt1a1, P-gp, Mrp2, Mrp2 and the relevant Oatp transporters.   

13.    As the authors have not published further details on the used bioanalytics, the authors need to provide detailed data on the method validation. Especially data on inter- and intra-day accuracy and precision as well as stability and matrix effects should be provided. This is current standard according to guidelines from FDA and EMA. This should be part of the Supplement.

14.    The authors used the student’s t-test for their analysis. However, this test assumes standard distribution which is mostly not given in animal experiments. It is suggested to use the non-parametric Mann Whitney test. Finally, multiple testing (e.g. for the transporter expression data) is so far not considered.

15.    The PK data in Table 2 do not indicate any significant differences. Please reanalyze and indicate.

16. Some Figures need statistical analysis (e.g. Figure 11).

17.    To support the study hypothesis, please provide hepatic and serum levels of the investigated bile acids.

17.    Please explain the finding that ezetimibe exposure was markedly increased only by oral but not intravenous administration in CCl4-treated animals. How can the authors exclude influences of CCl4 on intestinal Ugt1a1 or intestinal transporters such as P-gp or Mrp2/3?